# Different genetic mechanisms mediate spontaneous versus UVR-induced malignant melanoma

Blake Ferguson[1], Herlina Y Handoko[1], Pamela Mukhopadhyay[1], Arash Chitsazan[1], Lois Balmer[2,3], Grant Morahan[2], Graeme J Walker[1†]*

[1]Drug Discovery Group, QIMR Berghofer Medical Research Institute, Herston, Australia; [2]Centre for Diabetes Research, Harry Perkins Institute of Medical Research, Perth, Australia; [3]School of Medical and Health Sciences, Edith Cowan University, Joondalup, Australia

**Abstract** Genetic variation conferring resistance and susceptibility to carcinogen-induced tumorigenesis is frequently studied in mice. We have now turned this idea to melanoma using the collaborative cross (CC), a resource of mouse strains designed to discover genes for complex diseases. We studied melanoma-prone transgenic progeny across seventy CC genetic backgrounds. We mapped a strong quantitative trait locus for rapid onset spontaneous melanoma onset to *Prkdc*, a gene involved in detection and repair of DNA damage. In contrast, rapid onset UVR-induced melanoma was linked to the ribosomal subunit gene *Rrp15*. Ribosome biogenesis was upregulated in skin shortly after UVR exposure. Mechanistically, variation in the 'usual suspects' by which UVR may exacerbate melanoma, defective DNA repair, melanocyte proliferation, or inflammatory cell infiltration, did not explain melanoma susceptibility or resistance across the CC. Instead, events occurring soon after exposure, such as dysregulation of ribosome function, which alters many aspects of cellular metabolism, may be important.
DOI: https://doi.org/10.7554/eLife.42424.001

*For correspondence:
Graeme.Walker@qimr.edu.au

Present address: †Experimental Dermatology Group, The University of Queensland Diamantina Institute, Woolloongabba, Australia

Competing interests: The authors declare that no competing interests exist.

## Introduction

Cutaneous malignant melanoma (MM) is well known to be associated with high levels of sun exposure. However, this is only true for intermittent rather than chronic exposures, with indoor workers having a higher risk for MM than outdoor workers (*Gandini et al., 2005*). Examples of intermittent sun exposures include number of waterside vacations and number of severe sunburns. One melanoma subtype, lentigo maligna melanoma (LMM), is invariably linked to chronic sun exposure. Individual risk for the development of MM is in part due to the number of UVR-induced mutations incurred (i.e. the environmental factor), but it is also due to genetic variation that controls skin color (degree of protection due to pigment), DNA repair capability (*DiGiovanna and Kraemer, 2012*), propensity to burn (inflammation), failure of programmed death of a damaged cell, and/or other factors that control melanocyte behavior (*Sample and He, 2018*). Genome-wide association studies (GWAS) point to genes regulating aspects of cell division control (e.g. *CDKN2A/MTAP*, *PLA2G6*, *TERT*), DNA repair (e.g. *PARP1*, *APEX1*, *ATM*), and pigmentation (e.g. *MC1R*, *ASIP*, *TYR*, *SLC45A2*, *TYR*) as the main players in conferring MM risk (*Gerstenblith et al., 2010*; *Duffy et al., 2018*). However only a small fraction of the variance in MM risk is explained by these genes, suggesting that there are still other genes involved in conferring MM risk in the general population (*Hulur et al., 2017*).

Clearly, MM is a heterogeneous disease with respect to both innate and somatic genetics, and also to environmental factors since each individual with MM was exposed to different levels of sun

**eLife digest** Melanoma is a type of skin cancer. Melanoma tumors form in the skin's pigment-producing cells or melanocytes. Growing evidence points to complex interactions between genetics and environmental exposures that contribute to the risk of developing melanoma. Ultraviolet (UV) radiation from the sun causes genetic mutations in melanocytes. This sun exposure interacts with genetic variations that may make people more or less vulnerable to such DNA damage. For example, genetic variations that control skin color or the cell's ability to repair DNA, and that influence how easily people develop sunburn, all affect whether UV damage leads to melanoma. However, some forms of melanoma are not caused by sun exposure at all.

Most of the genetic variations linked to melanoma have a small effect on the risk of developing the disease. So, it is unlikely that these genes alone cause melanoma. Few studies have been able to map the complex interactions between genes and the environment that lead to melanoma. So far, it has been unclear if there are different genetic mechanisms that lead to an increased risk for sun-exposure linked melanoma and non-sun linked melanoma.

Now, Ferguson et al. show that variations in the genes involved in DNA repair during normal cell growth are linked to non-sun linked melanoma. Sun-linked melanoma, on the other hand, was associated with genes involved in the production of proteins in part of the cell called ribosomes. In the experiments, the effects of both UV light and various genetic variations were assessed across many different strains of mice. Mutations that impair the cell's ability to repair UV-induced DNA damage or that contribute to excessive inflammation in response to sunburn did not increase melanoma susceptibility in these experiments.

Ferguson et al. show that the amount of UV-induced DNA damage alone does not explain melanoma risk, which may not always depend on skin pigmentation. The experiments also suggest that non-UV linked melanoma is caused by a different mechanism than sun exposure-linked melanoma. Learning more about different genetic factors that affect the risk of developing different types of melanoma may help scientists develop more specific treatments.

DOI: https://doi.org/10.7554/eLife.42424.002

exposure. The most common superficial spreading melanomas (SSM) are sometimes found on sun-exposed body sites, but more often on non-sun exposed sites, and there can be great diversity in terms of the number of UVR signature mutations in individual lesions (*Mukhopadhyay et al., 2017*). While the number of UVR signature mutations in MMs strongly suggests a role for UVR in MM, the number of mutations present in a MM is due to not just obvious levels of exposure, and protective and repair differences between individuals, but also other factors, for instance the type of exposure. There is in vivo evidence that less intense exposures may be more important for skin cancer induction than more intense doses which cause more apoptosis (*Lan et al., 2016*), and even for a single exposure there are significant differences in skin responses between the same dose (i.e. the same levels of DNA damage) administered with high intensity over a short period versus a low intensity for a longer period (*Iida et al., 2016*). Our skin can protect itself via pigmentation responses (tanning), but also by 'photo-adaption', which is independent of pigmentation levels (*Palmer et al., 2006*). Thus, the skin of different individuals can respond and adapt to various forms of sun exposure in different ways, and there are potentially multiple and interacting mechanisms which might explain how UVR exposure could initiate or accelerate MM development in the general population.

For the above reasons, most experimental work on UVR carcinogenesis has used animal models. Natural genetic variation can confer resistance to many cancer types in mice (*Balmain, 2002*), and it is of great interest to determine why this is so. Most mouse MM models rapidly develop tumours after neonatal UVR exposure. Such models have provided tractable experimental systems to determine a MM action spectrum (*De Fabo et al., 2004*), to assess which type of UVR-induced DNA adducts are required, and to study the role of UVR-induced DNA damage, inflammation, and immunosuppression (*Walker, 2008*). It is not yet clearly known why a single neonatal UVR exposure so efficiently accelerates MM onset. We have shown previously that it is not via the acquisition of unrepaired UVR-induced damage leading to mutations in important cancer genes (*Mukhopadhyay et al., 2016*). A number of factors have been proposed to play a role: 1) there is a

muted inflammatory response to UVR in neonates associated with immunosuppression (*Wolnicka-Glubisz et al., 2007*); 15,16), *Muller et al., 2008*) there is a heightened sensitivity of neonatal melanocytes to proliferate following UVR (Walker et al., 209) that is driven by inflammatory cytokines, especially interferon-γ (*Zaidi et al., 2011*).

We used the $Cdk4^{R24C}::Tyr\text{-}NRAS^{Q61K}$ (hereafter termed *Cdk4::NRAS*) mouse as a UVR-induced MM model. Somatic *NRAS* mutation is carried by 27% of MMs (http://www.cbioportal.org), and in ~90% of MMs the p16/CDK4/pRb pathway is deregulated via mutations in *CDKN2A*, *CDK4*, or *RB1*, and/or *CDK4* or *CCND1* amplification (*Sheppard and McArthur, 2013*). We studied the development of MM in mice of 70 diverse genetic backgrounds carrying these transgenes. To do so, we utilized the Collaborative Cross (CC), a set of recombinant inbred mouse strains generated from eight original founder strains, designed to enable rapid gene mapping (*Churchill et al., 2004*; *Morahan et al., 2008*). The CC is ideal for systematic analysis studies to discover modifier genes for complex diseases. Mice from each inbred CC strain may be considered as 'clones' of each other. Related to the CC is the diversity outbred (DO) population, in which mice descended from the same eight founders are generated as outbred stock (*Churchill et al., 2012*). The DO system allows very high levels of heterozygosity and recombination of CC founder alleles, but each DO mouse is genetically unique and not reproducible for experimentation requiring testing of multiple mice. The CC system has allowed us to study the influence of germline genetic background on MM induction using experimentally controlled UVR exposures. This approach tries to explain UVR-induced MM susceptibility and resistance by integrating the complex interaction of many kinds of genetic and biological information, and as such should provide much more realistic insights into MM than simple disease models focusing on single genes or proteins in isolation (e.g. *Hamilton and Yu, 2012*).

## Results

### Assessment of NRAS<sup>Q61K</sup> and BRAF<sup>V600E</sup> transgenics as models for UVR-induced melanoma

Before embarking on the screen for melanoma modifier genes in mice, we assessed whether there may be better murine models to work with. All models tested were on the FVB strain background. Given that $BRAF^{V600E}$ mutation is more common than $NRAS^{Q61K}$ in MM overall, we studied the inducible $Braf^{V600E}$ model developed by the MacMahon lab (*Dankort et al., 2009*) combined with the knock-in mutant $Cdk4^{R24C}$ mouse. $Cdk4^{R24C/R24C}::Tyr\text{-}Cre^{ER}::Braf^{V600E}$ mice were studied in three groups. In one group, the spontaneous MM group, Braf was induced by topical tamoxifen (tam) at P1, P2, and P3. In the next group, we applied Tam at P1, P2, and P3, then exposed the mice to a single neonatal UVB dose at post-natal day 3 (P3) (*Figure 1A*). For the final group, we first exposed to UVR at P3, then treated with Tam at P7, P8 and P9 (*Figure 1B*). Surprisingly, we saw no significant difference in MM age of onset between any cohort (*Figure 1C*). Melanoma is not observed in $Cdk4^{R24C/R24C}$ mice without carrying a melanocyte-specific Ras pathway mutation, with or without neonatal UVR (*Hacker et al., 2006*), showing that in our experiments with the $Cdk4^{R24C/R24C}::Tyr\text{-}Cre^{ER}::Braf^{V600E}$ model, $Braf^{V600E}$ must have been induced by the tamox application. In contrast, using the $Cdk4^{R24C/R24C}::Tyr\text{-}NRAS^{Q61K}$ model (*Ferguson et al., 2010*) the single neonatal UVR exposure significantly accelerated MM age of onset (*Figure 1D*).

As another context in which to assess the role of the engineered mutation in mouse models of UVR-induced MM, we studied the $Trp53^{F/F}::Tyr\text{-}Cre(ER)::Tyr\text{-}NRAS$ model in which the *Trp53* deletion is induced by tamox application (26) (*Figure 1E*), whereas in these mice the $NRAS^{Q61K}$ mutation is not inducible, so is present through development. Tamox treatment (i.e. *Trp53* deletion in melanocytes) accelerated both spontaneous and UVR-induced MM. But there was no difference in MM onset whether or not *Trp53* was deleted before or after neonatal UVR. Thus for both $Braf^{V600E}$ and *Trp53*-inducible models the ability of neonatal UVR to accelerate MM may not be dependent upon whether the engineered mutation is present in melanocytes at the time of UVR exposure. Instead, it may be due to a more generalized effect via differences in DNA repair, melanocyte number and proliferative response, or inflammatory response, as has been outlined previously (*Mukhopadhyay et al., 2016*; *Walker et al., 2009*; *Zaidi et al., 2011*). But an oncogenic mutation in melanocytes seems to be a prerequisite. In sum, the *Cdk4::NRAS* model was best suited for breeding with CC mice to look for QTLs associated with UVR-dependent MM.

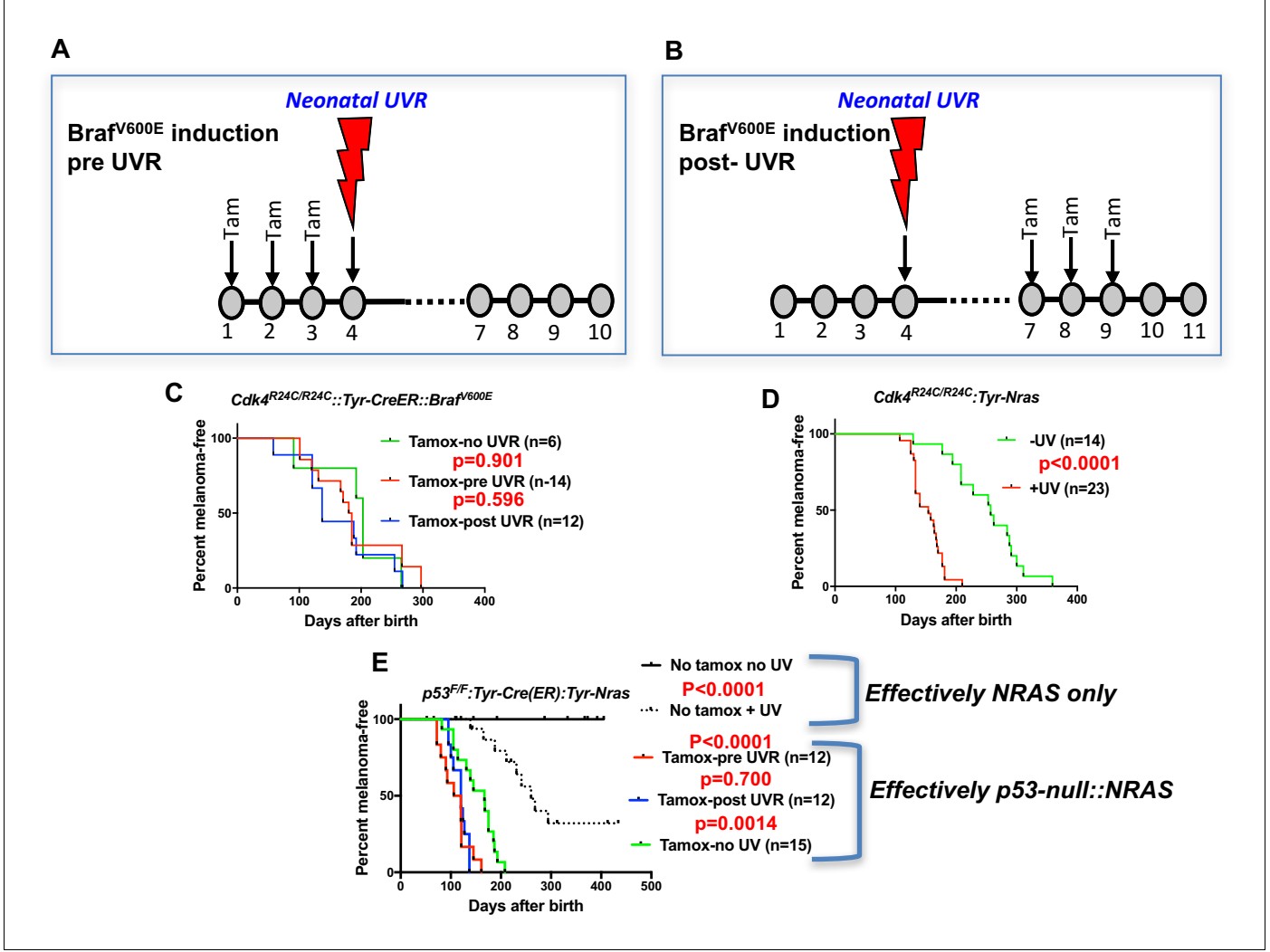

**Figure 1.** UVR-induced melanoma induction in different transgenic models. Schematic representation of timing of Tamoxifen application for induction of BRAF$^{V600E}$. Tam = tamoxifen (**A**) before UVR exposure, (**B**) after UVR exposure. (**C–E**) Comparison of UVR-induced MM-free survival between genotypes. Kaplan-Meier curves show the time to spontaneous and UVR-induced MM development. The age of onset (days after birth) was defined by the appearance of the first melanoma. Animals that died without developing MM were censored. (**C**) $Cdk4^{R24C/R24C}::BRAF^{V600E}$ with mutation induction before and after neonatal UVR, (**D**) $Cdk4^{R24C/R24C}::Tyt-NRAS^{Q61K}$ mice, (**E**) $Trp53^{F/F}::Tyr-Cre::TyrNRAS^{Q61K}$. Here we have included non tamox-treated animals which carry the effective genotype of *Tyr-NRAS* only, as shown to the right of the graph. Green, blue, and red lines show melanoma-free survival after various timings of tamox treatment. We aimed to study at least 20 mice in each group. 20 animals per group is sufficient to detect a difference in penetrance of 40% with statistical power of 80%.

DOI: https://doi.org/10.7554/eLife.42424.003

## QTLs for spontaneous melanoma age of onset in *Cdk4::Tyr-NRAS* mice

We tested 38 CC strains to discover QTLs that modify median age of onset of spontaneous MM per strain in CC X *Cdk4::Tyr-NRAS* progeny (*Ferguson et al., 2015*). The phenotype was encoded as median age of MM onset per strain (*Figure 2B*), and genetic analyses performed using the Gene Miner platform (*Ram and Morahan, 2017*). This software uses a logistic regression matrix model over the reconstructed haplotypes matrix to produce genome-wide distribution of P values (ANOVA chi-squared). We used a false discovery rate of p<0.001 to define significant genome wide linkage. We identified a major effect QTL on mouse chromosome (chr) 16. The -Log$_{10}$(P) −1 interval = 14.8–21.4 megabases (Mb), a region containing 45 genes (*Figure 2C*). Examination of the founder haplotype coefficients in the significantly linked interval on chromosome 16 showed that the causal variant for early age of onset of MM was derived from the 129/SvJ founder (hereafter termed 129S). We

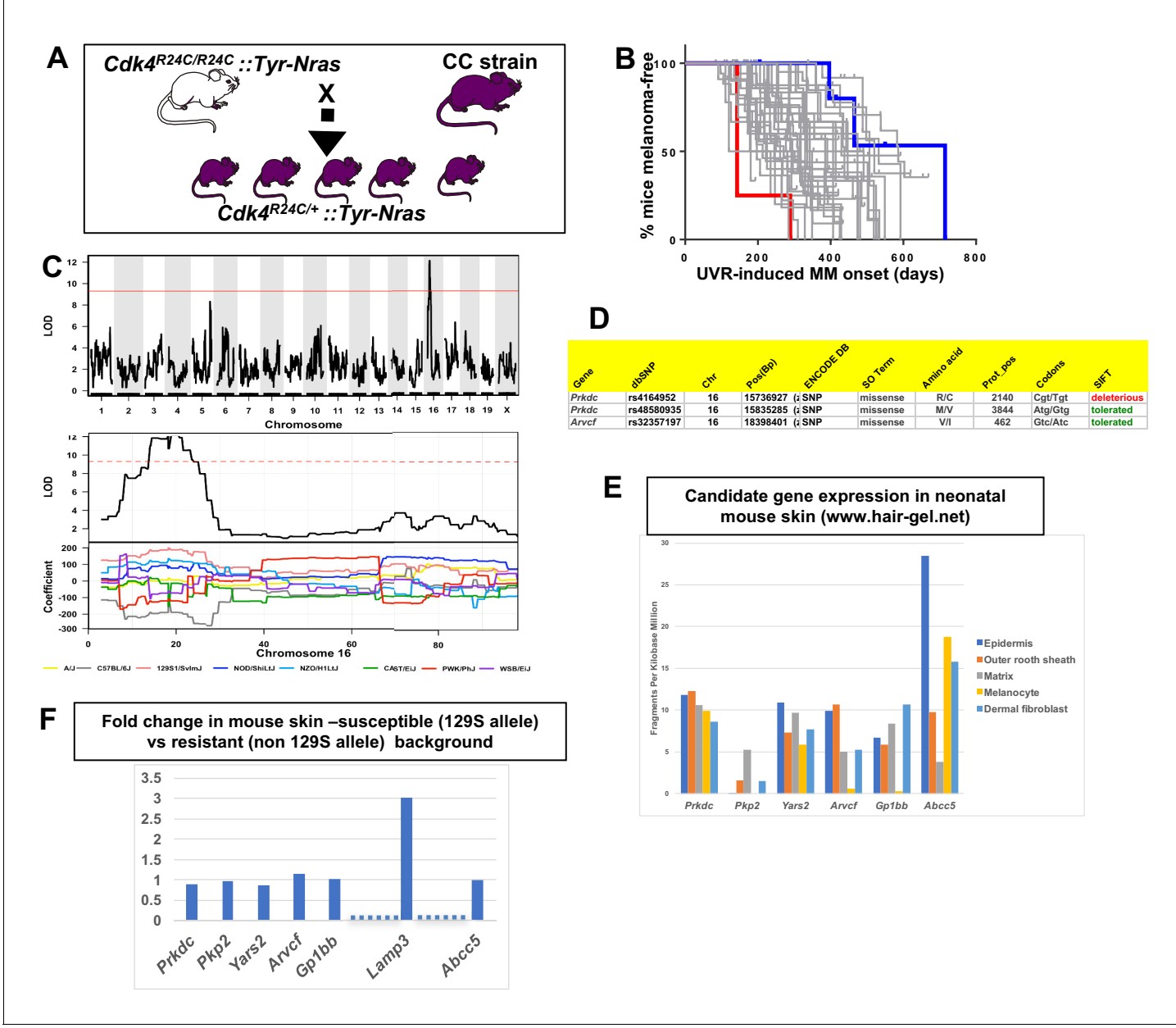

**Figure 2.** Mapping genes for spontaneous melanoma. (**A**) Schematic representation of breeding protocol to generate mice carrying 50% of their genome from the relevant CC strain. (**B**) Kaplan-Meier MM free survival curve showing age of onset on MM for all 38 strains. Strain in red shows he fastest age of onset. Note that there are four mice in the cohort, each onset value represents two mice. Blue shows slowest median age of onset. (**C**) Top panel shows genome-wide scan based on spontaneous MM age of onset in 38 CC strains. The average age of melanoma onset per strain was based upon onset for at least five mice for every strain. Significance of differences between groups calculated using the Log rank (Mantel Cox) test. Genotyping, construction of CC strain haplotypes, and linkage analysis was performed as described in *Ram and Morahan (2017)*. The x-axis shows the chromosomal position and the y-axis shows the 2log10(P) values; the P-values were derived from the linkage haplotype data. Bottom panel shows plot of LOD scores along chromosome 16, with plot of the calculated log-odds ratio of eight founder alleles over the chromosome where the founders are color-coded. (**C**) Genes within the −1 -Log10(P) interval carrying putative protein changing mutations. (**D**) Genes containing potentially regulatory intronic, 5'or 3' UTR variants. (**E**) Expression level of genes in the HAIR-GEL skin gene expression database (28). Y axis denotes FPKM (Fragments Per Kilobase Million). (**F**) Gene expression fold changes in mouse skin from 129S (which carries the susceptibility allele) compared to C57BL/6, NOD, and FVB, which do not. Based on gene expression values from RNA sequencing. *Lamp3* is separated from the other genes since it is the only one in the figure not located near a 129S-specific regulatory SNP defined in the ENCODE database.

DOI: https://doi.org/10.7554/eLife.42424.004

ascertained which genes within the interval were the best candidates by cataloguing DNA variants that are carried on the causal 129S haplotype, and not any of the other founder haplotypes. Mining the Sanger Mouse Genomes and ENCODE databases for 129S-specific variants revealed eight candidates, two of which (*Prkdc* and *Arvcf*) carry non-synonymous mutations (*Figure 2D*), while all 8 (*Prkdc, Pkp2, Yars2, Arvcf, Gp1bb,* and *Abcc5*) had 129S-specific single nucleotide potentially regulatory polymorphisms (SNPs) in their 5' or 3' UTR, or introns. It is possible that one of the 129S-specific SNPs near the candidates may regulate another gene elsewhere in the genome, or that there are other 129S-specific regulatory SNPs nearby any of the 45 genes in the region that were not detected by ENCODE.

Next, we looked at skin gene expression of the six candidates. They are expressed in neonatal mouse skin and/or hair follicle, including in melanocytes, as published in the Hair Gel database (*Sennett et al., 2015*) (*Figure 2E*). We then performed global gene expression analysis (RNAseq) (*Supplementary file 1*) of skin from adult mice from various laboratory strains (AJ, NOD, B6, FVB, and 129S) (*Figure 2F*), four of which are CC founders, for which the genome sequences are available in the Sanger database We found no differences between susceptible (129S allele-carrying) and resistant strains (non 129S allele- carrying; AJ, NOD, FVB, B6) that would help demarcate a candidate(s). Only one gene (*Lamp3*, lysosomal-associated membrane protein 3) of the potential expressed sequences within the region was significantly differentially expressed between the susceptible and resistant strains assessed (*Figure 2F*), but the nearest 129S-specific regulatory SNP is located 10 Kb away, near *Gp1bb*, and the next about 500 Kb away, near *Avcrf*. Unfortunately, there is no skin expression quantitative trait locus (eQTL) data for mouse skin, and it was not possible to consult the GTEX eQTL database for eQTLs in human skin since the *LAMP3, GP1BB,* and *ACVRF* genes in humans are scattered on different chromosomes.

Therefore, we hypothesized that a functional effect on phenotype was most likely the presence of missense mutations rather than regulatory expression changes on the causal allele. *Prkdc* carried two missense mutations, one defined as deleterious by SIFT. This is the same *Prkdc* (R2140C) mutation previously mapped and validated as a modifier of lymphomagenesis (*Mori et al., 2001*), breast cancer (*Yu et al., 2001*), and adenoma associated with ionizing radiation (*Degg et al., 2003*). PRKDC (also known as DNA-PKc) plays a critical role in ensuring genome integrity and in cancer in general by mediating ligation of double stranded breaks in DNA (*Goodwin et al., 2015*). The other gene within the interval that is a likely candidate is *Arvcf* (armadillo repeat gene deleted in velo-cardio-facial syndrome), a catenin protein family member. This family plays an important role in the formation of adherens complexes, which are thought to facilitate communication between the inside and outside environments of a cell. This carries on the causal 129S allele a missense mutation that is defined as tolerated by SIFT, so is perhaps not as likely to be the causal gene. In sum, while we cannot in effect rule out another gene within the region, the most likely candidate is *Prkdc*, since it carries a missense mutation on the causal allele shown before to confer cancer susceptibility, with *Acvrf* and *Lamp3* arguably less likely candidates.

We noted that other genes involved in sensing DNA damage and repairing double stranded breaks (e.g. *PARP1, ATM, APEX1*) are in linkage disequilibrium with SNPs associated with melanoma risk in GWAS (*Hulur et al., 2017*). We reasoned that even though *PRKDC* is not a GWAS hit, we could determine whether variation in its expression is correlated with expression of these GWAS genes. We examined global gene expression in non-sun exposed human skin across the GTEx cohort (Genenetwork.org). Networks of the top 500 genes correlated with *PRKDC* were constructed at a confidence value of 0.9 using STRING (https://string-db.org). The most significant network for molecular function was RNA binding (p=$1.22^{-37}$ false discovery rate), and in KEGG pathways DNA replication (p=$1.22 \times 10^{-8}$). *PARP1* (at 170, r = 0.46, p=$2 \times 10^{-16}$), and *APEX1* (at 287, r = 0.43, p=$8 \times 10^{-14}$) were in the top 300 most significantly correlated genes with *PRKDC*. *ATM* was not in the top 500, but its relative *ATR* was at number 105 (r = 0.49, p=$4 \times 10^{-16}$). While these correlations are based only on gene expression across the GTEx cohort, not any other aspect of gene function, they do point to the possibility that *PRKDC* is associated with pathways associated with DNA double strand break repair, components of which are encoded by other genes which confer MM risk in the general population.

## QTLs for UVR-induced melanoma age of onset in *Cdk4::Tyr-NRAS* mice

CC-transgenic progeny strains from 70 CC strains were exposed to a single neonatal exposure then followed until MM developed (*Figure 3A*). Median age of melanoma onset per strain was scored as the phenotype (*Figure 3B*). Only lesions developing on the UVR-exposed dorsal surface (where the overwhelming majority developed) were counted. We identified a major effect QTL on mouse chr.1 (-Log$_{10}$(P) −1 interval = 187.8–189.2 Mb) (*Figure 3C*), a region containing 10 genes. We ascertained which genes within the interval were the best candidates by cataloguing DNA variants on the causal allele that vary between susceptible or resistant strains. The causal allele was carried by AJ and NOD. Mining the Sanger Mouse Genomes and ENCODE databases for NOD/AJ-specific variants revealed four candidates carrying variants specific to the causal allele (*Tgfb2, Rrp15, Spata17, and Gpatch2*). *Rrp15* was the only one carrying missense mutations (*Figure 3D*), while the other three genes (*Tgfb2, Spata17*, and *Gpatch2*), had AJ/NOD-specific single nucleotide polymorphisms (SNPs) in their 5' or 3' UTR, or introns (*Figure 3E*).

Next we looked at the Hair Gel database to determine whether these genes were expressed in skin. All but *Spata17* were, essentially ruling this gene out as a candidate (*Figure 3F*). We then looked at skin gene expression (*Supplementary file 1*) between AJ, NOD, and FVB, which carry the Chr. one allele, and B6, DBA, and 129S which do not (*Figure 3G*). There were no significantly differentially expressed genes between groups, suggesting a missense rather than a regulatory causal variant. Thus, while we cannot rule out *Tgfb2* and *Gpatch2*, by those criteria *Rrp15* is the best candidate. The causal allele (AJ/NOD) carried two *Rrp15* missense variants, both defined by SIFT as likely to be 'tolerated'. But one at amino acid 117 is most likely to be the causal mutation given that it is a Glu > Gln change, whereas the other is Ala > Val, which is likely to be silent. *Rrp15* encodes a ribosomal subunit that is part of pre 40S and pre 60S subunits that is important for rRNA transcription and ribosome biogenesis. Furthermore, *Rrp15* knockout in vitro causes nucleolar stress by activating the Mdm2-Trp53 axis, and subsequently a G1-S phase cell cycle blockage (*De Marchis et al., 2005*; *Dong et al., 2017*). Deregulation of the ribosome complex decreases the fidelity and patterns of mRNA translation and many other downstream events (*Quin et al., 2014*; *Pelletier et al., 2018*). Although much of the effect on cell behaviour is via altering the stabilisation of p53 by Mdm2, this can also occur via p53-independent mechanisms (*James et al., 2014*).

## Confirmation of the chromosome one locus using diversity outbred (DO) mice

DO mice harbor frequent recombinations throughout their genomes. We hypothesized that some DO animals (*Churchill et al., 2012*) would have informative recombinants of the causal AJ/NOD allele to make them useful for fine mapping the linked chr.1 interval. We tested DNA from 314 DO mice, initially using six polymorphic markers across an 8 Mb region around *Rrp15*. We selected a subgroup of 8 DO mice with apparent recombinations within the AJ/NOD alleles across the region and genotyped them using a panel of 60 SNPs (20 of which are the most informative and shown in *Supplementary file 2*. SNPs were chosen based on their ability to discriminate the AJ/NOD allele from the other eight founders, especially those variants private for AJ, NOD, or both. While this NOD/AJ allele was scattered throughout the region in different DO mice, as expected we observed many recombinations across the chr.1 region of interest. However there were many gaps and alleles for which we could not unambiguously call the founder haplotypes on both strands across the region (*Supplementary file 2*). To adequately do this, many more SNPs would have to be tested. Nonetheless we crossed each of the eight selected DO mice with *Cdk4::NRAS* transgenics, and studied UVR-induced MM onset in the progeny (*Figure 3H*). Because of the limitations of our SNP-based map of the region, to genotype the DO mice we used manual Sanger sequencing to genotype more densely, in particular in the region of our candidate gene *Rrp15* (*Figure 3I*). As seen in *Figure 3H*, the predicted causal SNPs in *Rrp15* do not segregate with fast MM onset as they did in the CC strains. We hypothesized that lack of penetrance of the *Rrp15* allele was due to the introduction of additional resistance alleles elsewhere in the genome due to the high levels of recombinations and heterozygosity in the DO genomes. We tried to circumvent this by backcrossing transgenic-DO mice onto C57BL/6. Mice from several litters were followed, and after two backcrosses we assessed UVR-induced MM age of onset, with progeny genotyped immediately around *Rrp15* by Sanger sequencing. We found that the penetrance of the causal *Rrp15* variant was restored on backcrossing: mice

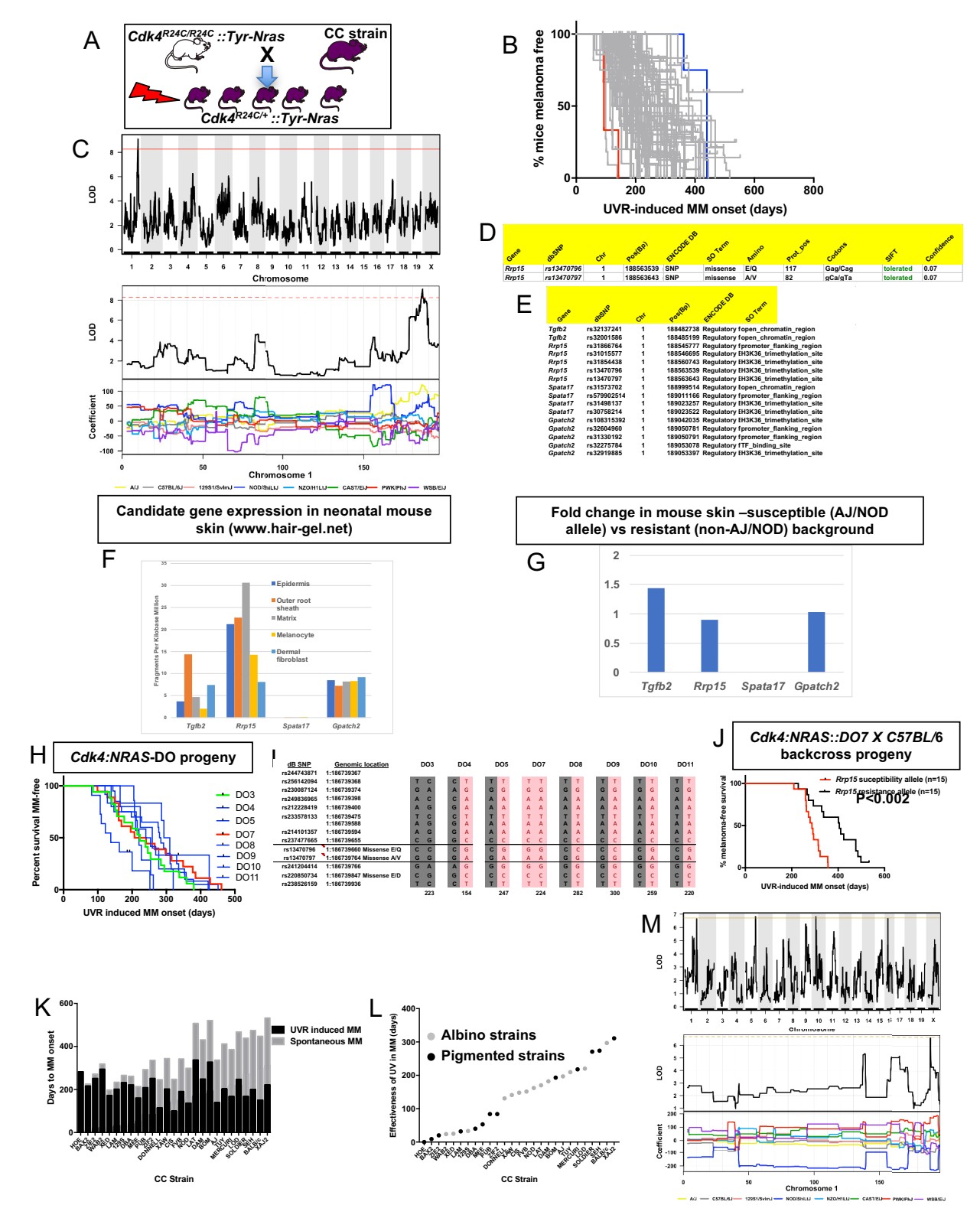

**Figure 3.** Mapping genes for UVR-induced melanoma. (**A**) Schematic representation of breeding protocol to generate mice CC progeny. Pups were exposed to a single UVR exposure at post-natal day 3. (**B**) Kaplan-Meier MM free survival curve showing age of onset on MM for all 38 strains. Strain in red shows the fastest age of onset. There are two data points (four mice) for this strain cohort, but each onset value represents two mice that developed MM at the same time. Blue shows slowest median age of onset. (**C**) Top panel shows genome-wide scan based on neonatal UVR-induced

*Figure 3 continued*

MM age of onset in 70 CC strains. The average age of melanoma onset per strain was based upon onset for at least five mice for every strain. Significance of differences between groups calculated using the Log rank (Mantel Cox) test. Genotyping, construction of CC strain haplotypes, and linkage analysis was performed as described in *Ram and Morahan (2017)*. The x-axis shows the chromosomal position and the y-axis shows the 2log10 (P) values; the P-values were derived from the linkage haplotype data. Bottom panel shows plot of LOD scores along chromosome 1, with plot of the calculated log-odds ratio of eight founder alleles over the chromosome where the founders are colour-coded. (D) Genes within the −1 -Log10(P) interval carrying putative protein changing mutations. (E) Genes containing potentially regulatory intronic, 5'or 3' UTR variants. (F) Expression level of genes in the HAIR-GEL skin gene expression database (28). Y axis denotes FPKM (Fragments Per Kilobase Million). (G) Gene expression fold changes in mouse skin from susceptible (AJ, NOD, FVB) compared to resistant (C57BL/6, 129S) strains. Based on gene expression values from RNA sequencing of the skin. (H) Kaplan-Meier curve for UVR-induced MM free survival for *Cdk4::NRAS::DO* progeny for each of 8 DO strains. Red = homozygous for the AJ/NOD susceptibility allele at the *Rrp15* locus, blue-heterozygous, and green strains do not carry the allele. (I), Haplotypes for the parental DO mice at the *Rrp15* locus used to predict progeny genotype in *Figure 3H*. Pink boxes with re text = *Rrp15* susceptibility allele. Dark boxes denote the *Rrp15* resistance allele. (J) Kaplan-Meier curve for UVR-induced MM age of onset for *Cdk4::NRAS::DO7* progeny after two backcrosses onto C57BL/6. P-value calculated using the Log Rank test. (K) Plot depicting relationship between average age of onset spontaneous and UVR-induced MM for a subgroup of CC strains. (L) Plot depicting 'effectiveness' of UVR in exacerbating melanoma (spontaneous minus UVR-induced MM) onset from each strain. Grey dots - albino strains, black dots - pigmented strains. (M) Top panel shows genome-wide scan based on neonatal UVR 'effectiveness' in inducing MM. Bottom panel shows plot of LOD scores along chromosome 1, with plot of the calculated log-odds ratio of eight founder alleles over the chromosome where the founders are colour-coded. The brown line denotes 'suggested' linkage (FDR < 0.01), whereas an FDR < 0.001 is defined as significant linkage.
DOI: https://doi.org/10.7554/eLife.42424.005

carrying the NOD/AJ alleles of *Rrp15* had significantly earlier MM onset (*Figure 3J*). Thus, the propensity for neonatal UVR to accelerate melanoma is highly dependent upon genetic background influences. In addition, there are additional potential resistance alleles which can, if present, interact with the *Rrp15* susceptibility allele.

## Neonatal UVR acceleration of melanoma age of onset is dependent upon innate genetics

We further analyzed the role of neonatal UVR in accelerating melanoma age of onset by subtracting the average age of onset of spontaneous from that of UVR-accelerated melanoma (*Figure 3K*). This provides us with a further phenotype: the strain-specific rate of acceleration of MM onset by neonatal UVR (*Figure 3L*). The median age-of-onset for UVR-induced MM per strain was subtracted from the median onset for spontaneous MM, to denote what we have termed the 'effectiveness' of UVR in inducing MM. In resistant CC strains, neonatal UVR does not accelerate MM onset at all (e.g. HOE, BAX2), or does so by a very small amount, whereas for susceptible strains MM onset was accelerated by more than 300 days (e.g. SEH, XAJ2) as seen in *Figure 3K and L*. The phenotype is independent of pigmentation status (albino vs pigmented) (*Figure 3L*). The genome scan for this trait is shown in *Figure 3M*. Unfortunately, we only have both spontaneous and UVR-induced onset data for 27 strains. While this does not provide enough power to detect genome-wide significant linkage to this phenotype, there was a peak of suggestive significance at chromosome 1 p, overlapping with the QTL detected for UVR-induced age of onset, containing the *Rrp15* gene, with the NOD allele showing a different coefficient from the other strains.

## Skin gene expression changes after neonatal UVR

The way by which neonatal UVR accelerates MM may provide insights into early events in the initiation of this neoplasm. To search for pathways deregulated in neonatal UVR-exposed skin, we performed global gene expression chip studies with Illumina Beadchips on UVR exposed and non-exposed epidermis of wild-type FVB mice at various time-points (*Supplementary file 3*). The top 500 significantly up- and down- regulated genes after neonatal UVR were used to construct networks for UVR-induced gene expression changes at a confidence value of 0.9 using the STRING resource (https://string-db.org) (*Figure 4*). At 6 hr after UVR exposure, one can see strong significant evidence of ribosome biogenesis occurring, with downregulation of various metabolic pathways compared to control untreated skin (*Supplementary file 4*). At 10 hr, post-UVR metabolic activity is still suppressed, but DNA replication is either already occurring to some extent, or about to occur for repair and reconstruction of damaged cells. Upregulated p53 signaling was also observed. At 24 hr post UVR various pathways involved in cell proliferation (ribosomes and translation, metabolism) and

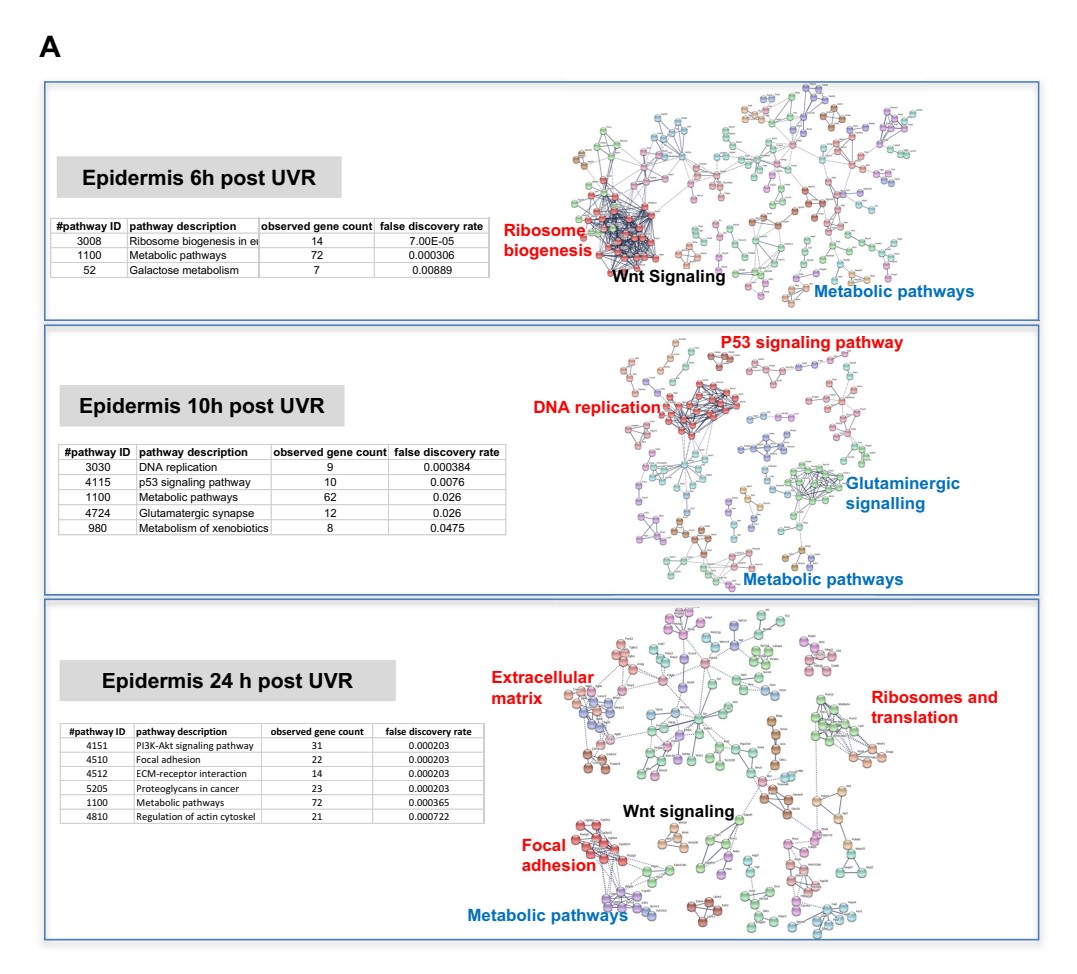

**Figure 4.** Gene expression networks induced by UVR. (**A**) Gene network analysis showing at various time-points after neonatal UVR the top 500 genes (all p<0.05) deregulated in UVR-treated neonatal epidermis versus untreated (from the same mouse on the side blocked with electrical tape). Based on three mice (control) and three mice (+UV) per time-point. Pathways labelled in red were upregulated after UVR, and those in blue downregulated. For those labelled in black some genes were up and some downregulated.

DOI: https://doi.org/10.7554/eLife.42424.006

reconstruction (extracellular matrix, metabolism) are significantly activated in the epidermis. One of the problems with using whole tissue is that one cannot discriminate in terms of cell type. As a way to put the skin gene expression into some perspective we wanted to look at skin gene expression changes at some days after UVR, since it is known that this period corresponds with an influx of immune cells into the dermis (*Zaidi et al., 2011*) and the influx of melanocytes into the epidermis (*Walker et al., 2009*). Therefore, we separately studied gene expression in the neonatal epidermis and dermis harvested at 3d after UVR (*Figure 5A*). There was a striking loss of immune markers in the epidermis, mostly reflecting UVR-induced migration of epidermal Langerhans cells to the dermis as expected. We also saw a strong gene expression cluster for genes involved in melanogenesis, reflecting the presence of melanocytes in the UVR-exposed epidermis, but not in control skin at the time-point 3 d after UVR. In the dermis, we observed networks reflecting increased cellular activity, in particular upregulation of kinases and interferon-induced enzymes. But in contrast to the epidermis, in the dermis we also observed a signal for myeloid cells, presumably reflecting the influx of macrophages, which occurs maximally at around this time post UVR (*Handoko et al., 2013*).

The critical question with respect to genes deregulated by neonatal UVR is the behavior of our candidate genes, one of which must be causal in accelerating MM. *Rrp15* was the only one of the chr.1 candidates significantly changed in expression by UVR (*Figure 5B*). Hence in addition to the

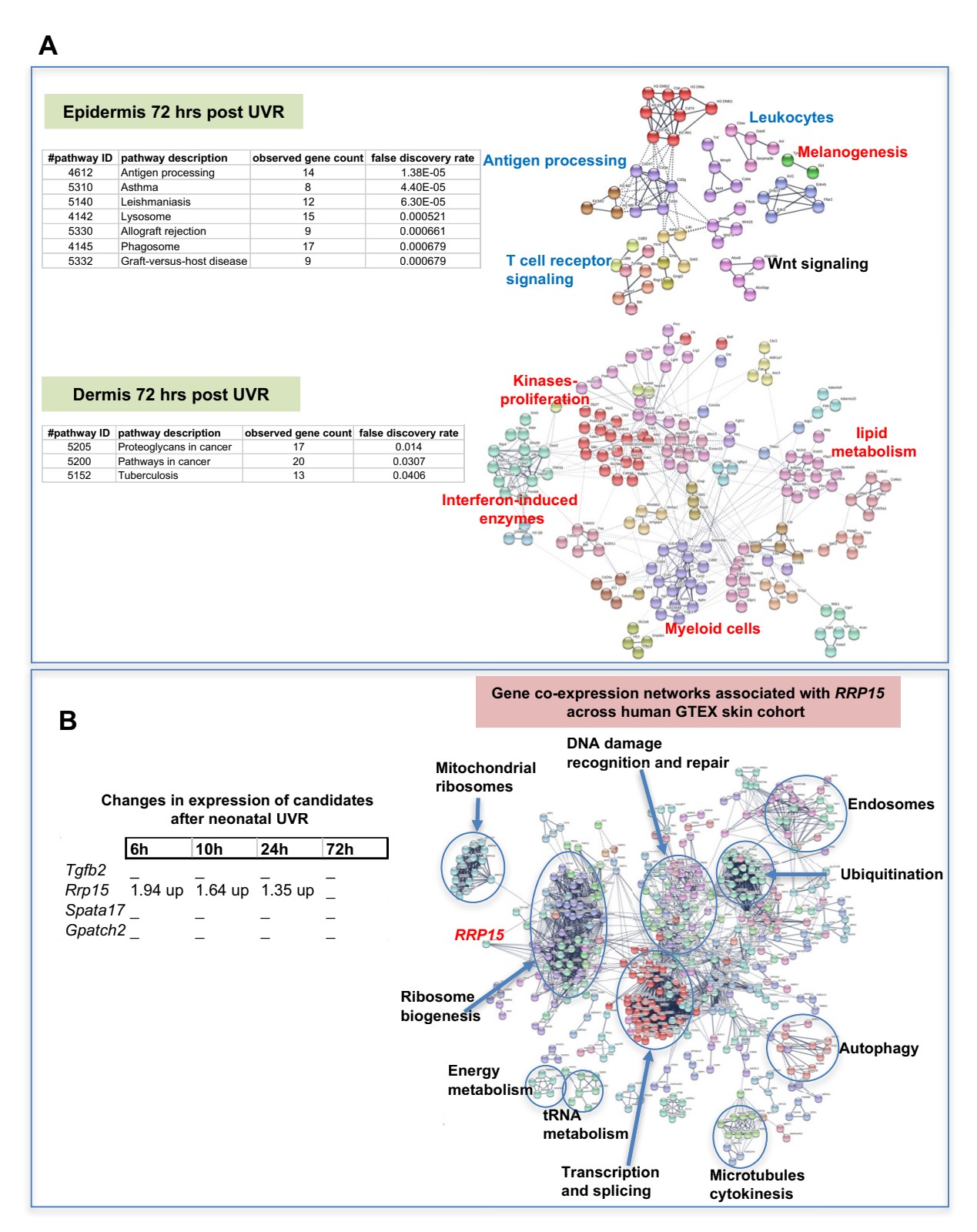

**Figure 5.** Gene expression networks induced by UVR. (**A**) Gene network analysis showing at epidermis (top panel), and dermis (bottom panel) 72 hr after neonatal UVR, the top 500 genes (all p<0.05) deregulated in UVR-treated neonatal epidermis versus untreated (from the same mouse on the side blocked with electrical tape). Pathways labelled in red were upregulated after UVR, and those in blue downregulated. For those labelled in black some genes were up and some downregulated. (**B**) Fold change of expression of candidate genes after neonatal UVR. The only gene within the candidate

*Figure 5 continued on next page*

*Figure 5 continued*

region on chr.1 changed after neonatal UVR was *Rrp15*. To the right is a network of genes correlated with *RRP15* expression across the sun-exposed site human skin GTEx cohort (Genenetwork.org). Networks of the top 500 genes correlated with *RRP15* were constructed at a confidence value of 0.9 using STRING (https://string-db.org).

DOI: https://doi.org/10.7554/eLife.42424.007

susceptible CC strains carrying a missense mutation in the putative *Rrp15* causal allele, the fact that it is the only candidate that responds to neonatal UVR adds weight to it being the best candidate in the linked genomic region. Our gene expression analysis was performed on whole skin, epidermis, or dermis, rather than individual cell types, allowing signals from a number of cells types, for example keratinocytes, immune cells, and melanocytes, to be taken into account. Since *Rrp15* is expressed virtually ubiquitously in the skin we do not know whether its mode of action in accelerating UVR-induced melanoma would be cell intrinsic or extrinsic. But we note that in a study of UVR-treated cultured melanocytes, ribosome metabolism was the most significantly altered pathway at 6, 12, and 24 hr after UVR (*López et al., 2015*), as in our whole skin studies. In addition, to gain a better sense of changes in skin correlated with *RRP15* expression we constructed gene networks based on the top 500 genes correlated with *RRP15* expression across the sun-exposed anatomical site human skin GTEx cohort (*Figure 5B*). As expected changes in *RRP15* gene expression are correlated with ribosome biogenesis, but also many other aspects of cell behavior, including DNA damage recognition and repair, transcription, and splicing. Defects in any of these processes could in theory explain why MM is accelerated by UVR exposure not only in mouse strains which carry germline *Rrp15* variants, and putatively in humans also.

## Mechanism by which melanoma is accelerated by UVR

Since genetic background greatly influenced whether or not MM was accelerated by neonatal UVR, we studied various effects of UVR a few days after exposure on traits that may differ between susceptible and resistant strains. These included the rate of removal of UVR-induced cyclobutane pyrimidine dimers (CPDs), influx of inflammatory neutrophils and macrophages, and the proliferation of melanocytes (increase in melanocyte number in the epidermis) (*Figure 6A*). We again used readily-available laboratory strains (AJ, B6, DBA, FVB, NOD, 129 s), four of which are CC founders. For all six, we determined the median age of onset of MM after neonatal UVR. We chose time-points 1, 4, and 7 days post UVR. In C57BL/6 mice, CPDs are generally removed by d4 after UVR, certainly by d7. Neutrophils generally infiltrate the skin by d1 after UVR and numbers decrease by d4. Macrophages are maximally present at 4d (39). At 4d, melanocytes have migrated to the epidermis in response to UVR, macrophages are at their maximum number, roughly concurrent with maximum epidermal melanocyte density. By the d7 post UVR time-point, the skin is ostensibly returned to normal, but in some mouse strains there may be a delay in some measures.

First, across the six laboratory strains we found no differences in the capacity to remove CPDs, as measured by the proportion of skin cells carrying them post-UVR. However, there was great variation between the various strains in the size and timing of the melanocyte and myeloid cell responses after UVR (*Figure 6A*). To ascertain whether the level of any of these responses could be correlated with age of MM onset, we compared results for the lab strains stratified in terms of whether they carry the susceptible or resistant allele at chr.1 (e.g. around *Rrp15*) (*Figure 6B*). When compared in this way none of these commonly purported mechanisms by which UVR exacerbates MM were significantly associated with onset of UVR-induced MM, although we found a non-significant trend for the susceptible strains to show higher levels of neutrophil influx at d1 after UVR. As an additional assessment, we performed a correlation test between MM age of onset and the phenotypes measured and found no significant correlation, although d1 neutrophil infiltration was nearest to significance.

If one accepts that the low number of strains compared (3 vs 3) makes it difficult to attain statistical significance for a mechanism with a complex milleiu of events, one could argue that neutrophil influx may have an effect on exacerbating MM. Thus we set out to determine whether depletion of neutrophil infiltration using a neutrophil-blocking anti-Ly6g antibody would influence subsequent melanoma induction in *Cdk4::NRAS* mice. We injected blocking antibody at 1d before UVR, the day of UVR, and 2d after UVR. First we assessed neutrophil depletion after three injections by taking skin

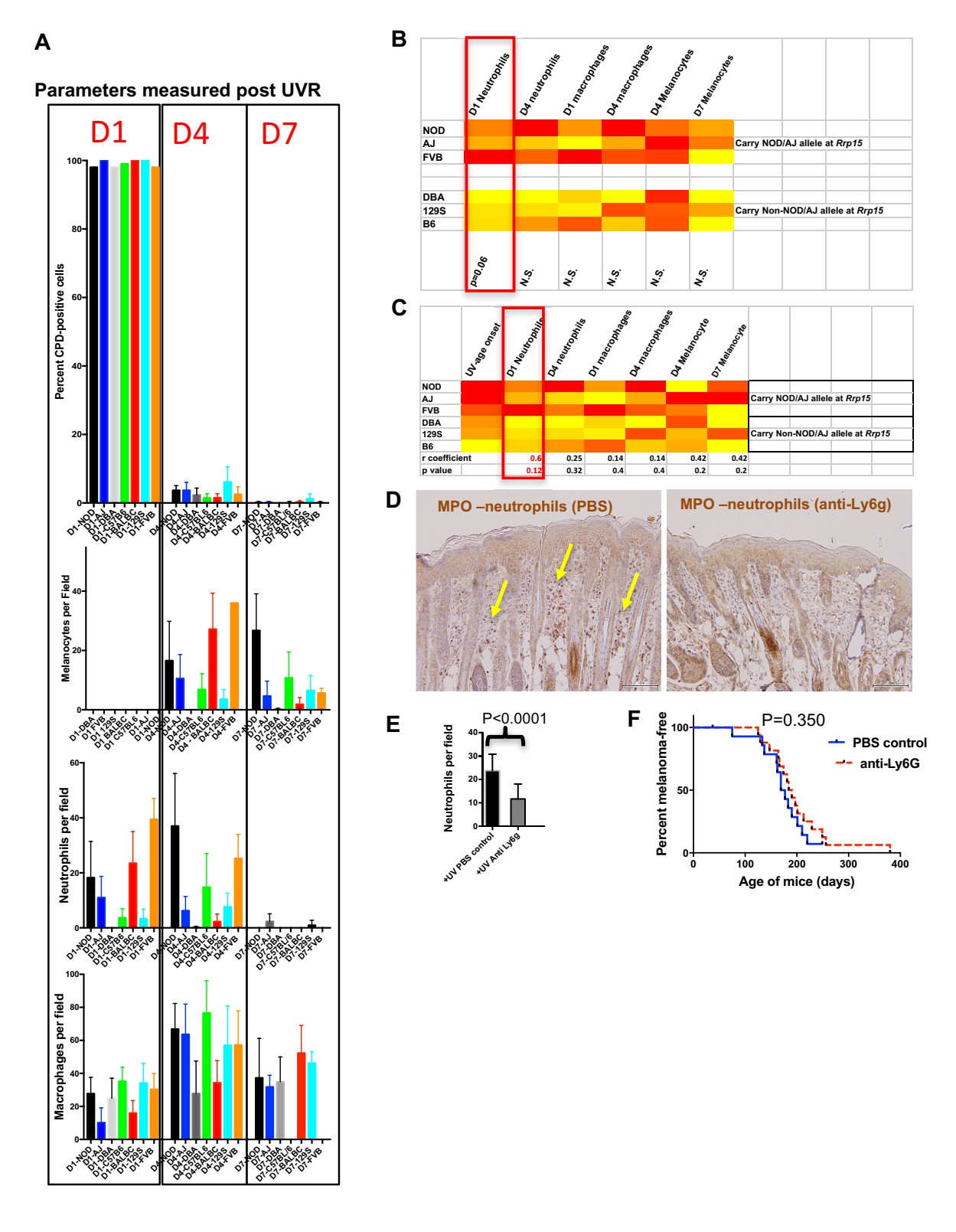

**Figure 6.** Immune and melanocyte proliferative responses post UVR in different mouse strains. (**A**) Parameters measured post UVR in different mouse strains. Top bar graph shows number of cells (stained with anti-CPD antibody) positive for pyrimidine dimers at 1 (D1), 4 (D4) and 7 (D7) in whole skin after neonatal UVR. Second bar graph shows the number of epidermal melanocytes (stained with anti-Sox10 antibody) per field. Third graph shows the number of dermal neutrophils (stained with anti-myeloperoxidase) per field. Fourth graph shows the number of dermal macrophages (stained with anti-

*Figure 6 continued on next page*

*Figure 6 continued*

F4/80 antibody) per field. For each replicate time-point cells were counted from at least three mice, with at least 10 fields counted per mouse. (B) Levels of each measured parameter post UVR presented as a heatmap. Strains are separated according to whether they carry the *Rrp15* susceptibility allele (NOD, AJ, FVB) or not (DBA, 129S, (B6). For each parameter, groups were treated separately and analysed for significant differences between groups using the Mann-Whitney U test using PRISM. (C) Heatmap using the same data as in the previous heatmap, but this time strains are listed according to age of onset of melanoma (fastest to slowest), and each parameter analysed for Pearson correlation co-efficient with age of onset of melanoma using PRISM. P-value for a two-tailed test based on 95% confidence interval. Final row shows correlation Pearson r value and p value for the correlation. (D) Both panels show myeloperoxidase (MPO) staining of neonatal FVB mouse skin at 24 hr post UVR. Yellow arrows denote neutrophils, which were present in PBS-treated skin but not skin treated with neutrophil depleting Ly6g antibody. (E) Graph shows average number of neutrophils per field in UVR treated skin with (n = 17) or without (n = 15) treatment with neutrophil depleting antibody. p-value calculated using the Mann Whitney U test. Mice were collected from over a number of litters, with each litter divided in two to randomly establish treatment and control groups. (F) Kaplan-Meier analysis of melanoma free survival in neonatal UVR-treated mice. We aimed to study at least 20 mice in each group. 20 animals per group is sufficient to detect a difference in penetrance of 40% with statistical power of 80%. Actual numbers analysed n = 18 for the neutrophil-depleted group, and n = 15 for the control group. Significance of differences between groups calculated using the Log rank (Mantel Cox) test. There was no significant difference in melanoma age of onset whether or not neonates were treated with neutrophil depleting Ly6g antibody.

DOI: https://doi.org/10.7554/eLife.42424.008

sections and staining with neutrophil-specific anti-myeloperoxidase antibody (*Figure 6C*). Anti-Ly6g treatment significantly depleted neutrophil influx after neonatal UVR (*Figure 6D*). We exposed two cohorts of *Cdk4::NRAS* mice to anti-Ly6g anti-neutrophil depleting Ab, or PBS control respectively, then performed a longitudinal study for MM age of onset, but saw no difference in the age of onset of MM between cohorts (*Figure 6E*). Thus, inhibition of the neutrophil influx did not suppress UVR-induced genesis.

## Discussion

There are likely to be multiple and interacting mechanisms which might explain how UVR exposure could initiate or accelerate different types of MM. We have used a mouse MM model to study the development of MM on many CC strain backgrounds. The *Cdk4::NRAS* model we used is a well-characterized model of UVR-induced MM, and in terms of an acceleration of MM onset after neonatal UVR it behaves similarly to other models which carry constitutive oncogenic mutations in melanocytes (e.g. the *Mt-Hgf* model) (*Noonan et al., 2001*). But, surprisingly, we found that MM was not accelerated in a model carrying an inducible $Braf^{V600E}$ mutations in its melanocytes. It is known that modes of engineered oncogene expression in melanocytes can influence other phenotypes also. For example, $Braf^{V600E}$ induced in melanocytes *in utero* is incompatible with embryonic viability (e.g. *Dankort et al., 2009*), whereas in a non-inducible transgenic $Braf^{V600E}$ models it is not (*Wurm et al., 2012*). We do not know why MM was not accelerated in the inducible $Braf^{V600E}$ model, but one possibility is that there may be simply more activated melanocytes in the neonatal $Tyr-NRAS^{Q61K}$ trans-genics at the time of UVR exposure since $NRAS^{Q61K}$ is expressed *in utero*, whereas $Braf^{V600E}$ is induced only for a short period of 3 days before UVR (i.e. P1-P3). We favour such an explanation rather than a mechanistic difference between the respective oncogenes (*Braf* vs *NRAS*) per se, since in another similar inducible $Braf^{V600E}$ model repeated UVR exposures significantly accelerate MM when the oncogene is activated at 4 weeks of age (*Goel et al., 2009*). In a model using inducible *Trp53* deletion instead of constitutive $Cdk4^{R24C}$ we found that UVR accelerated MM, but whether *Trp53* was deleted before or after UVR was inconsequential. This could be somewhat akin to the work of the Evans laboratory who showed that although TP53 is very important in helping modulate the acute DNA damage response to radiation, its function in this period does not contribute to protecting against transformation, in fact it is only within the long lag phase leading to cancer development that TP53 acts as a tumour suppressor (*Christophorou et al., 2006*). Therefore, mouse models appear to differ with respect to whether and by how much neonatal UVR accelerates MM. There is no perfect animal model for 'generalized' MM, but one would expect that the *Cdk4::NRAS* trans-genic in combination with the CC should provide useful information as to how genetic background can influence UVR-induced murine MM.

Age of onset of spontaneous MM across the CC was underpinned by germline variation in the *Prkdc* gene on chr.16. This is overwhelmingly the best candidate in the mapped interval as the mutation carried on the 129S susceptibility allele has been functionally validated in other mouse models

of cancer (*Yu et al., 2001*; *Degg et al., 2003*; *Goodwin et al., 2015*). Although it is known to be involved in recognition and removal of DNA damage, it is unclear whether its function in MM would be cell intrinsic or extrinsic, since it is ubiquitously expressed in skin cell types. In addition, this is quantitative genetics, and although the QTL explains a large proportion of the phenotypic variation, other cooperating loci may also exist, for instance on chr.5, (*Figure 3C*), which might become statistically significant (or disappear) if more CC strains were tested. In terms of possible human relevance, other genes (e.g. *PARP1*, *APEX1*) involved similarly in DNA repair, and whose expression is correlated with that of *PRKDC* in human skin, are associated with population-based MM risk (*Duffy et al., 2018*). Thus *PRKDC* may be an example of a modifier of MM in mice, but mediated by other genes of the same pathway in humans. But the mechanistic result of *Prkdc* deregulation might be similar to the effects of deregulating MM risk genes like *PARP1* and *APEX1*, that is, all ultimately influencing DNA repair. Hence our genetic screen appears to have revealed mechanistically relevant information, notwithstanding that the particular causal gene variants can be different between mouse and man. One can also consider *Prkdc* as a MM modifier gene, that is, our genetic screen detects MM modifiers in the context of a mouse model which carries a germline *Cdk4* variant. In keeping with this, *PRKDC* was also one of the few DNA repair genes associated that modified MM risk in MM-prone families (*Liang et al., 2012*), many of which carry a *CDKN2A* mutation, and a few *CDK4* mutation. If we assume that *Cdk4::NRAS* mice may in some respects be ostensibly a model for familial MM, the findings of *Liang et al., 2012* further support the possible relevance of our findings in human disease.

One of the most striking findings from our study was that the genetic polymorphisms modifying MM onset were very different between spontaneous and UV-induced disease (*Figure 7*). This is particularly notable since in this model MM is accelerated by just a single UVR exposure. The age of onset of neonatal UVR-induced MM in the transgenic-CC progeny across the CC was linked to a chr.1 locus containing a strong candidate, *Rrp15*, with a missense mutation. Taken together with the fact that of the candidates within the linked region, only *Rrp15* was significantly differentially expressed between susceptible and resistant strains, only *Rrp15* was upregulated in the epidermis of

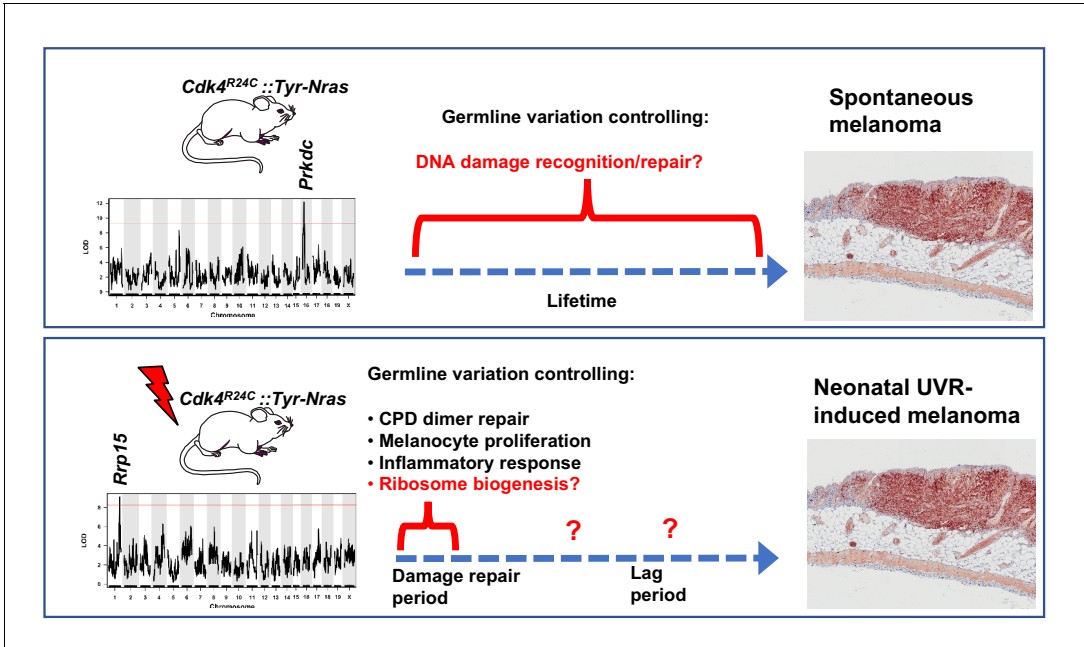

**Figure 7.** Schematic representation showing candidate genes within QTLs regulating melanoma age of onset in mice. Figure also shows putative mechanisms which accelerate melanoma in either the spontaneous or UVR-induced contexts. Germline gene variation influencing double strand break recognition and repair probably throughout the life of the animals controls spontaneous MM development. Germline gene variation influencing ribosome function and protein synthesis either during the acute damage repair period after neonatal UVR, or perhaps during the lag period leading to tumour initiation, explain acceleration of MM by UVR.
DOI: https://doi.org/10.7554/eLife.42424.009

neonatal skin after UVR exposure (at 24 hr), we deemed it the strongest candidate. In addition, gene ontology and KEGG pathway analysis of UVR-induced gene expression changes in neonatal mouse skin revealed that ribosome biogenesis is one of the major gene networks upregulated at 6 and 24 hr after UVR. One caveat here is that for the UVR-induced gene expression studies we used whole epidermis (or dermis). Notwithstanding that there are differences in melanocyte and immune cell density over time between 1 and 3 d after neonatal UVR, most of the signal would have come from keratinocytes at all time- points since they are the most numerous cells in the epidermis. But our analysis of UVR-induced gene expression changes, based upon fold-change, are sensitive enough to pick up gene networks that reflect predictable changes in melanocyte and immune cell density (although not the type of immune cells). *Rrp15* is expressed in all cell types within the neonatal skin (*Figure 3F*), but we do not know whether its mode of action in modifying melanocyte transformation is cell autonomous or non-cell autonomous. We assume that the upregulation of ribosomal metabolism we observe at 24 hr after UVR reflects mostly changes in keratinocytes, but very similar responses have also been found in UVR-treated cultured melanocytes (*López et al., 2015*). Single cell sequencing of different skin cells may help resolve these problems, but even such methods also have their drawbacks including unwanted gene expression changes induced soon after removal of cells from their microenvironment (*Yuan et al., 2017*).

We have used inferences taken from a variety of sources, which are by themselves not confirmatory, to build a strong case for a role for variation in *Rrp15* function influencing the propensity for UVR to accelerate MM. There is some evidence that SNPs near *RRP15* are also associated with human MM in some contexts. The world melanoma genetics consortium found such evidence using a method for candidate genetic association to detect variants that may not reach genome-wide statistical significance after correction for multiple testing (*Schoof et al., 2012*; *Wurm et al., 2012*). There was no information on individual sun exposure in the tested cohorts. Of the 39 immune-related genes tested, SNPs near *LGAL3* and *TGFB2* were the most significantly associated with MM risk (*Rendleman et al., 2013*). Of note, the *RRP15* and *TGFB2* genes are located adjacent to one another and just 7 kb apart. Which of the two is the causal gene is difficult to elucidate in the human genetics study, but our systems analysis work using the CC has allowed us to build a very strong case for *Rrp15*.

We discovered that genetic background dramatically influences the propensity for melanocyte transformation after UVR. As well as discovering a gene likely to be associated with this, we also examined differences in skin responses between susceptible versus resistant mouse strains. We did not observe major differences in the propensity for removal of UVR-induced CPDs, nor in melanocyte proliferation, nor macrophage influx. We observed a weak correlation with the number of skin-infiltrating neutrophils at d1 after neonatal UVR and UVR-induced MM onset. However, depleting neutrophils before and after neonatal UVR did not reduce the time of onset of MM, somewhat in line with our previous finding that depletion of macrophage infiltration also did not abrogate the MM-accelerating effect of UVR (*Handoko et al., 2015*). It could be argued that by using immunohistochemical staining to assess removal of UVR-induced CPDs we could miss subtle differences between mouse strains in the repair process that may be consequential in terms of leaving a molecular memory of UVR damage in the transformed cells. But even our exome sequencing on the neonatal UVR-induced mouse melanomas does not suggest a major role for UVR-induced mutagenesis (*Mukhopadhyay et al., 2016*).

Researchers tend to look for measurable skin responses to UVR exposure to try to understand how melanocytes may be destabilized and ultimately transformed by DNA damage. The 'usual suspects' include defective DNA repair, photo-immunosuppression, inflammation, and cell proliferation, which are observable in the days following exposure. These can all enhance carcinogenesis in specific contexts. But our results suggest that earlier events in the few hours post UVR such as aberrant ribosome activity that can cause inappropriate protein expression, may be more important. It is not clear whether this would be acting during the acute damage repair period after neonatal UVR, or during the lag period leading to tumour initiation. The notion of a particular constellation of gene networks that vary between CC strains and confer resistance to the MM-accelerating effects of UVR may not be dissimilar to what occurs in amphibians, where regenerating limbs, but not non-regenerating body parts, are resistant to carcinogen-induced cancer (*Sarig and Tzahor, 2017*), despite the fact that both anatomical sites incur the requisite DNA damage. Particular molecular networks within skin cells in the MM-resistant strains appear to work against transformation, despite animals from all

strains being exposed to high levels of UVR-induced damage. Hence in keeping with the fact that even non-cancerous human skin can carry UVR signature mutations in cancer genes (*Martincorena et al., 2015*), incorrectly repaired UVR-induced DNA damage leading to somatic mutation is in essence necessary but not sufficient to exacerbate MM, and the presence of germline variants for melanoma susceptibility and resistance is very important. Biological systems (e.g. skin and skin UVR responses) are very complex and varied across a population of different individuals. Our systems analysis strategy has attempted to harness such complexity and in doing so has resulted in the discovery of some potentially important findings with regards UVR-induced melanoma. We have performed a genetic screen for natural genes regulating MM age of onset and found surprisingly that different genes mediate spontaneous and UVR-induced MM susceptibility (*Figure 7*). We have identified a strong candidate genes and potential mechanisms in both cases.

# Materials and methods

### Key resources table

| Reagent type (species) or resource | Designation | Source or reference | Identifiers | Additional information |
|---|---|---|---|---|
| Gene (*Mus musculus*) | Cdk4-R24C | PMID:11606789 | | crossed onto FVB/N background |
| Gene (*Homo sapiens*) | NRAS-Q61K | PMID:11606789 | | crossed onto FVB/N background |
| Gene (*M. musculus*) | p53F/F | PubMed: 10783170 | 008361 - B6; 129S4-Trp53<tm5Tyj>/J - The Jackson Laboratory | crossed onto FVB/N background |
| Gene (*M. musculus*) | Braf-V600E | PMID: 17299132 | 017837 - B6.129P2(Cg)-Braf<tm1Mmcm>/J - The Jackson Laboratory | crossed onto FVB/N background |
| Gene (*M. musculus*) | Tyr-Cre ER | 16676322 | 012328 - B6.Cg-Tg(Tyr-cre/ERT2)13Bos/J - The Jackson Laboratory | crossed onto FVB/N background |
| Genetic reagent (*M. musculus*) | Collaborative Cross resource | Geniad Pty Ltd, and the Animal Resource Centre (ARC), Western Australia | | Mice descended from eight founders generated as recombinant inbred stock |
| Genetic reagent (*M. musculus*) | Diversity Outbred mouse resource | Geniad Pty Ltd, and the Animal Resource Centre(ARC), Western Australia | | Mice descended from eight founders generated as outbred stock |
| Biological sample (*M. musculus*) | skin samples from mouse subjects | Animal Resource Centre(ARC), Western Australia | | C57B6, NOD, A/J, 129s mouse strains |
| Antibody | anti F4/80 from CD68 rat monoclonal antibody | Abcam | Abcam 6640, CI: A3-1 (Cambridge,UK) | for macrophages. 1:400 dilution |
| Antibody | anti Sox10 goat polyclonal | Santa Cruz Biotechnologies | SC-17342 (N-20) Santa Cruz Biotechnology (Dallas Tx, USA) | for melanocytes. 1:200 dilution |
| Antibody | Monoclonal anti-thymine dimer, Clone H3 | Sigma Aldrich | Anti-Thymine dimer Clone H3 (T1192), Sigma Aldrich (St Louis, MO, USA) | for Cyclobutane pyrmidine dimers. 1:400 dilution |
| Antibody | Anti-Ly6G Rat anti-neutrophil monoclonal antibody | Abcam | Anti-Ly6G rat monoclonal antibody Abcam ab2557: Clone NIMP-R14 (Cambridge, UK) | for Neutrophil staining (1:400 dilution) and depletion (undiluted) |
| Antibody | Anti-MPO | Abnova | Abnova rabbit anti-MPO Clone 14328 (Taipei, Taiwan) | for Neutrophils after neutrophil depletion 1:100) |

*Continued on next page*

*Continued*

| Reagent type (species) or resource | Designation | Source or reference | Identifiers | Additional information |
|---|---|---|---|---|
| Sequence-based reagent | Vector NovaRed Peroxidase Substrate kit | Vector Laboratories | SK-4800, Vector Laboratories (CA, USA) | |
| Commercial assay or kit | Mouse expression array | Illumina | Illumina TotalPrep RNA amplification, Illumina MouseWG-6 v2.0 Expression Beadchips | |
| Commercial assay or kit | RNA sequencing | Illumina | Illumina mRNA kit, Illumina HiSeq with 50bp single reads | RNA-seq samples mapped to mouse genome MM10 using TopHat2 |
| Commercial assay or kit | RNA isolation kit | Qiagen | Rneasy Kit | |
| Software, algorithm | Geneminer platform | Geneminer | http://130.95.9.22/Geniad2/ | Ram R, Morahan G. Using Systems Genetics to Understanding the Etiology of Complex Disease. Methods Mol Biol 2017;1488:597-606. |
| Other | UVB lamps for irradiation | Phillips | 6 lamps TL100W 12RS UVB lamps | |

## Mouse melanoma model

$Cdk4^{R24C/R24C}::Tyr\text{-}NRAS^{Q61K/+}$ mice are previously described in *Ferguson et al. (2010)*. We crossed $Cdk4^{R24C/R24C}::Tyr\text{-}NRAS^{Q61K/+}$ mice with breeding partners from each CC strain (*Ferguson et al., 2015*). Hence all study mice are $Cdk4^{R24C/+}::Tyr\text{-}NRAS^{Q61K/+}$. All experiments were undertaken with institute animal ethics approval (A98004M). Mice were sacrificed before tumors exceeded 10 mm in diameter. In some melanoma-resistant strains lymphomas developed in some mice at >400 days of age. Such mice were counted as melanoma-free at the age of death. Each phenotypic measurement is based upon at least 4 and up to 15 mice per CC strain background. $p53^{F/F}$ mice (carrying floxed alleles allowing Cre-mediated excision of exons 1–10) were obtained from the Mouse Models for Cancer Consortium (http://mouse.ncifcrf.gov). Melanocyte-specific *Trp53* deletion in $p53^{F/F}$/*Tyr-Cre (ER)/Tyr-NRAS* mice was induced via topical application of 8-OH-tamoxifen (15 mg/ml in DMSO) at P0, 1 and 2. For the studies involving the BRAF model we used the inducible $BRAF^{V600E}$ model generated by (24). All these mice were of FVB strain background.

## CC breeding

Collaborative cross mice were obtained from the Animal Research Council (ARC) in collaboration with Prof Grant Morahan of the Harry Perkins Institute of Medical Research, Perth, Australia. A/J, C57BL/6J (B6), 129/SvJ (129S), DBA, and NOD/ShiLtJ (NOD) were purchased from the Animal Resources Centre, Western Australia.

## UVR treatments

Pups (3-day-old) were exposed to a single UVB exposure from a bank of 6 Phillips TL100W 12RS UVB lamps (Total UVB dose, 5.9 kJ/m$^2$, or an erythemally-weighted dose of 1.8 kJ/m$^2$) UVB.

## Phenotypic characterization

We have previously described a system for visual tracking of lesions developing on the FVB $Cdk4^{R24C/R24C}::Tyr\text{-}NRAS^{Q61K/+}$ mice and a histology-based staging system (*Wurm et al., 2012*) and mice on the various CC strain backgrounds were scored in this way. Briefly, lesions were excised after death followed by conventional histopathologic work-up with haematoxylin and eosin (H and E) staining. Each lesion was viewed and confirmed individually as MM by BF and GJW as described in Wurm et al. (42). Where there was any doubt about diagnosis, tumors were stained with Trp1 and/or Sox10. No skin tumors were observed apart from melanomas. We included in our analyses

melanomas that developed on dorsal surface only. Mice were sacrificed before tumors exceeded 10 mm in diameter.

## QTL analysis

The construction of the CC founder haplotypes is described in *Ram and Morahan (2017)*. For mapping we used a logistic regression matrix model over the reconstructed haplotypes matrix to produce genome-wide distribution of P values (ANOVA chi-squared). We used a false discovery rate of p=0.0001 to define significant genome wide linkage.

## SNP genotyping

We used a custom SNP approach performed by AGRF, a custom array of 92 SNPs across the 180–190 Mb region of mouse Chr. 1. Genotyping was performed using the Sequenom mass array system.

## Sanger sequencing

Target fragments chosen to contain multiple SNPs were PCR-amplified, then cleaned from excess primers and nucleotides using CleanSweep PCR Purification (Applied Biosystems, Life Technologies, Carlsbad, CA, USA). Sequencing was carried out using BigDye Terminator v3.1 Cycle Sequencing and then run on ABI Prism DNA Sequencers (Applied Biosystems). The sequencing traces were compared using Multiple SeqDoc chromatogram comparison programme (http://research.imb.uq.edu.au/seqdoc/multi.html).

## Mouse expression array

Five hundred ng of total RNA from each tumor was used as the starting material to produce cRNA, following Illumina TotalPrep RNA Amplification protocol. From each sample, 1500 ng of cRNA were hybridised to Illumina MouseWG-6 v2.0 Expression BeadChips (Illumina, San Diego, CA, USA) and then scanned. Data were extracted using Genome Studio (Illumina, San Diego, CA, USA) and then imported into GeneSpring GX 11.5.1 (Agilent Technologies, Santa Clara, CA, USA), before subsequent analysis. Normalisation was performed using the R package LUMI, and differential analysis done using LIMMA. Multiple testing correction was carried out using the Benjamini-Hochberg procedure. Differentially expressed genes with p-adjusted values < 0.05 were considered significant.

## RNA seq

Skin gene expression experiments were undertaken with institute animal ethics approval (A98004M). RNA was isolated using RNeasy Kit and libraries generated with Illumina mRNA kit. Sequencing was performed using Ilumina HiSeq chemistry, with 50 bp single reads. RNA-seq samples were mapped to mouse genome MM10 using STAR. Quality control metrics were computed using RNA_SeQC version 1.1.18, and expression values estimated using RSEM version 1.2.30. We corrected for library size by dividing each sample's count by millions of reads mapped. We used the 'calcnormfactors' function from the EdgeR package to obtain TMM factors and used these to correct for differences in RNA composition.

## Gene expression correlation and gene network study

To generate interconnected networks based on correlations, gene lists were clustered using STRING (http://string-db.org/). STRING creates networks representing the best available knowledge of gene interconnections. Each protein-protein interaction is annotated with 'scores' indicating how likely an interaction should be true. Scores rank from 0 to 1, with one being the highest confidence. A score of 0.5 indicates roughly every second interaction might be erroneous. Gene-gene co-expression correlations were computed as Pearson product-moment correlations (r) in Genenetwork.org after removing outliers.

## Antibody staining

Skin from the pups at Day one after UVB radiation (D1), D4 and D7 was paraffin embedded. All immunohistochemistry staining was performed on pup skin sections (4 um) with standard DAB or NovaRed. Counts were performed on multiple fields from multiple skin sections from each of >3

mice (each field is ~1 mm in length). Macrophage staining used F4/80 from CD68 rat monoclonal antibody diluted to 1:400, Abcam ab6640, CI: A3-1 (Cambridge, UK). Melanocyte nuclear staining used Sox10 from sc-17342 goat polyclonal antibody (N-20) diluted to 1:200, Santa Cruz Biotechnology (Dallas, TX, USA). Cyclobutane pyrimidine dimers (CPD) staining was with monoclonal anti-thymine dimer, Clone H3 (T1192) diluted to 1:400, Sigma-Aldrich (St Louis, MO, USA). For this CPD staining, sections were first blocked in 1% hydrogen peroxide, then incubated in 50% ethanol, 30% ethanol/0.02N HCl, 0.05N HCl, and 0.07N NaOh/70% ethanol, before incubating in primary antibody. Neutrophils were stained using an anti-Ly6G as primary antibody, except after neutrophil depletion, when staining was done using MPO staining (see below). Anti-Ly6G was from rat anti-neutrophil monoclonal antibody Abcam ab2557: NIMP-R14 (Cambridge, UK) at a concentration of 1:100.

## Neutrophil depletion

Each *NOD/Cdk4$^{R24C/+}$::NRAS* pup was injected intraperitoneally, with either 100 ug *InVivo* MAb anti-mouse Ly6G (Bio C Cell, Beverly, MA, USA) or PBS (as a control) at P2, P3 and P5. At P3, all pups were also given UVR treatment. As the depletion was done using anti-Ly6G, staining to differentiate between depleted and non-depleted skin was done using Myeloperoxidase staining (Abnova rabbit anti-MPO, Taipei, Taiwan) at a concentration of 1:75.

## Statistical analysis

One-way ANOVA tests were used to determine the significant difference between means, using R. The survival of mice in each treatment group was estimated using Kaplan-Meier analysis (PRISM$^{TM}$), and the Log-Rank (Mantel-Cox) test was used to test for differences between the groups. For correlation comparisons Pearson correlation r value was calculated in PRISM along with the p-value for significant correlation.

## Acknowledgements

This work was supported by Melanoma Research Alliance, Washington DC.

## Additional information

### Funding

| Funder | Grant reference number | Author |
|---|---|---|
| Melanoma Research Alliance | Investigator Grant Award Number: 346859 2015-2018 | Graeme J Walker |

The funders had no role in study design, data collection and interpretation, or the decision to submit the work for publication.

### Author contributions

Blake Ferguson, Herlina Y Handoko, Conceptualization, Data curation, Formal analysis, Validation, Investigation, Methodology, Writing—review and editing; Pamela Mukhopadhyay, Data curation, Software, Formal analysis, Methodology, Writing—review and editing; Arash Chitsazan, Data curation, Investigation; Lois Balmer, Resources, Data curation, Methodology; Grant Morahan, Conceptualization, Resources, Formal analysis, Supervision, Funding acquisition, Methodology, Writing—review and editing; Graeme J Walker, Conceptualization, Resources, Data curation, Formal analysis, Supervision, Funding acquisition, Validation, Methodology, Writing—original draft, Project administration, Writing—review and editing

### Author ORCIDs

Blake Ferguson http://orcid.org/0000-0002-5643-6976
Graeme J Walker https://orcid.org/0000-0002-9392-8769

## Ethics

Animal experimentation: This study was performed in strict accordance with the recommendations Australian code of Practice for the care and use of animals for scientific purposes.. All of the animals were handled according to approved institutional animal care and use committee of the Queensland Institute of Medical research. The protocol was approved by the Committee (A98004M). No surgery was performed. Animals were sacrificed when tumours reached 10mm in diameter, or animals were otherwise distressed.

## Decision letter and Author response

Decision letter https://doi.org/10.7554/eLife.42424.018
Author response https://doi.org/10.7554/eLife.42424.019

# Additional files

## Supplementary files

• Supplementary file 1. Schematic representation of SNPs across the interval 4 Mb either side of *Rrp15*. Pink haplotypes represent the AJ/NOD allele, grey one or more of the other six founders, and clear regions were unable to be inferred. Results listed for 20 SNPs across 8 DO strains (DO1-DO11).
DOI: https://doi.org/10.7554/eLife.42424.010

• Supplementary file 2. List of normalized gene expression values for adult mouse skin. Unprocessed raw counts based on Fragments per kilo base per million mapped reads. Table shows results for whole skin from 8 weeks old mice of the following strains; AJ, NOD, 129S, FVB, and C57BL/6
DOI: https://doi.org/10.7554/eLife.42424.011

• Supplementary file 3. Fold change of genes at various time-points post neonatal UVR. Significantly up or down regulated genes based on analysis of either whole epidermis or whole dermis gene expression using Illumina MouseWG-6 v2.0 Expression BeadChips. At least three mice (control), and three mice (+UV) were used for each time-point.
DOI: https://doi.org/10.7554/eLife.42424.012

• Supplementary file 4. List of significant pathways upregulated in neonatal murine skin at various times after UVR. To generate interconnected networks based on correlations, gene lists were clustered using STRING (http://string-db.org/) based on the top 500 genes differentially under or overexpressed after neonatal UVR exposure. The terms are sorted by their enrichment value shown as statistically in terms of false discovery rate.
DOI: https://doi.org/10.7554/eLife.42424.013

• Transparent reporting form
DOI: https://doi.org/10.7554/eLife.42424.014

## Data availability

All data generated in this manuscript are provided in the manuscript and supporting files.

The following previously published dataset was used:

| Author(s) | Year | Dataset title | Dataset URL | Database and Identifier |
|---|---|---|---|---|
| GTEx consortium | 2018 | Skin -not sun exposed | http://gn2.genenetwork.org/show_trait?trait_id=ENSG00000162438.7&dataset=GTEXv5_Vag_0915 | GeneNetwork, phs000424.v5.p1 |

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
