## [Decision Letter]

Thank you for submitting your article "Different genetic mechanisms mediate spontaneous versus UVR-induced malignant melanoma" for consideration by *eLife*. Your article has been reviewed by three peer reviewers, and the evaluation has been overseen by a Reviewing Editor and Jeffrey Settleman as the Senior Editor. The reviewers have opted to remain anonymous.

The reviewers have discussed the reviews with one another and the Reviewing Editor has drafted this decision to help you prepare a revised submission.

Summary:

Overall, this manuscript addresses an important point regarding the etiology of melanoma in terms of UV versus non-UV related mechanisms. It makes excellent use of mouse genetics to identify QTLs associated with spontaneous versus UV generated melanoma. This has very important implications for the field, as it would suggest that different methods would be needed to target these two types of melanoma. Overall, all the reviewers were enthusiastic about the work and feel it makes an important contribution to the field. We would like the authors to address a few concerns, as detailed below, prior to publication.

Essential revisions:

1) Figure 1: have the tumors been tested to demonstrate that the expected mutations have been activated by Cre-mediated recombination? Since this is a somewhat unexpected result it would provide more confidence for the investigators to show that BRAF^V600E^ is expressed, and that p53 is knocked out as predicted.

2) One significant concern with the paper is the experiment on the skin gene expression changes after neonatal UVR. For this experiment, the gene expression was tested in skin, epidermis, and dermis whole tissues. Melanocytes only make up a very small fraction of skin (~1%); ~85% of skin is keratinocytes; and the immune infiltrate is expected to be highly heterogeneous. These factors make the gene expression analysis in the above whole tissues highly unreliable and indiscernible. It is not clear whether any useful conclusions can be made out of such gene expression analyses, especially in the absence of any validation studies. At a minimum, the authors should state the fraction of cells in the skin that are melanocytes versus other cells, or at least address this point in their Discussion.

---

## [Author Response]

Essential revisions:1) Figure 1: have the tumors been tested to demonstrate that the expected mutations have been activated by Cre-mediated recombination? Since this is a somewhat unexpected result it would provide more confidence for the investigators to show that BRAF^V600E^ is expressed, and that p53 is knocked out as predicted.

The reviewer has a good point. We have now included data and explanations in the manuscript which better show that the genes were included/deleted. However we did not test the tumours for expression of these genes, since they could be inadvertently switched off or lost by deletion in a tumour (this might particularly apply to p53). The proof that tamoxifen is deleting the respective genes instead comes from comparisons of the mice without tamox treatment, when the mice do not develop melanoma at all (in the case of Braf), or much slower, in the case of p53.

In terms of the Braf model, which carries the Cdk4 (constitutional) and Braf (melanocyte-specific) mutations, we have never seen a melanoma, with or without UVR, in *Cdk4^R24C^* mice without an accompanying melanocyte-specific Ras pathway mutation (Hacker et al., 2006, we have added this reference to the manuscript). Hence none of the Cdk4-Braf mice develop melanoma at all unless tamox is applied, and so Braf^V600E^ must be activated in the mouse melanocytes in our study. We have updated the text in the first paragraph of the Results section to explain this.

Likewise, with respect to the *p53floxed::TyrCre::Tyr-NRAS* mice, they develop melanoma much more rapidly after tamox treatment than without, showing that p53 must be knocked out. See Figure 1C, which now shows the age of onset of animals not treated with tamox (essentially NRAS only), compared to those treated with tamox (essentially *p53-null::NRAS*). This is shown for both spontaneous and UVR-induced melanoma, and the second paragraph of the Results section, along with Figure 1E, is now amended accordingly.

2) One significant concern with the paper is the experiment on the skin gene expression changes after neonatal UVR. For this experiment, the gene expression was tested in skin, epidermis, and dermis whole tissues. Melanocytes only make up a very small fraction of skin (~1%); ~85% of skin is keratinocytes; and the immune infiltrate is expected to be highly heterogeneous. These factors make the gene expression analysis in the above whole tissues highly unreliable and indiscernible. It is not clear whether any useful conclusions can be made out of such gene expression analyses, especially in the absence of any validation studies. At a minimum, the authors should state the fraction of cells in the skin that are melanocytes versus other cells, or at least address this point in their Discussion.

As suggested by the reviewer, it is a good idea to discuss the limitations, and we have significantly altered the third paragraph of the Discussion to point out the caveats with respect to whole tissue gene expression analysis in this context. In essence we explain that we do not know the mode of action of potential causal gene(s) such as *Rrp15*, i.e. is its action cell or non-cell autonomous, and that we have taken data from a variety of sources and experiments to build our case for it. We also add that there is no perfect method of gene expression analysis, and that even methods based upon gene expression analyses using disaggregated single cells also have their drawbacks. Hence we have to glean information on skin gene expression from where we can. Skin cell type quantitation: we now state in the manuscript that keratinocytes are by far the most common cell type in skin, but we did not attempt a quantitation since the proportion of keratinocytes could be higher after UVR-induced epidermal thickening, and certainly the number of melanocytes and immune cells changes greatly between 1 and 3 days after UVR.